# Changes in Myosin from Silver Carp (*Hypophthalmichthys molitrix*) under Microwave-Assisted Water Bath Heating on a Multiscale

**DOI:** 10.3390/foods11081071

**Published:** 2022-04-07

**Authors:** Haihua Cong, He Lyu, Wenwen Liang, Ziwei Zhang, Xiaodong Chen

**Affiliations:** 1Key Laboratory of Aquatic Product Processing and Utilization of Liaoning Province, College of Food Science and Engineering, Dalian Ocean University, Dalian 116023, China; lww159457@163.com (W.L.); zw03160928@163.com (Z.Z.); 2Collaborative Innovation Center of Provincial and Ministerial Co-Construction for Marine Food Deep Processing, Dalian Polytechnic University, Dalian 116034, China; 3School of Chemical Sciences, University of Auckland, Private Bag 92019, Auckland 1142, New Zealand; hlyu015@aucklanduni.ac.nz; 4Huilly Pharmaceuticals Ltd., Suzhou 215000, China; 5School of Chemical and Environmental Engineering, College of Chemistry, Chemical Engineering and Material Science, Soochow University, Suzhou 215123, China

**Keywords:** silver carp (*Hypophthalmichthys molitrix*), myosin, heating, protein structure, protein aggregation behavior

## Abstract

To further prove the advantages of microwave-assisted water bath heating (MWH) in low-value fish processing, the effects of different heating methods (two heating stage method, high temperature section respectively using MWH1, MWH2, MWH3, WH—water heating, MH—microwave heating) on secondary and tertiary myosin structures, SDS-PAGE, surface morphology, scanning electron microscopy (SEM), and particle size distribution were compared and analyzed. The findings revealed that MH and MWH aided in the production of gel formations by promoting myosin aggregation. Myosin from silver carps demonstrated enhanced sulfhydryl group and surface hydrophobicity after MWH treatment, as well as a dense network structure. The distribution of micropores becomes more uniform when the microwave time is increased. Actually, the total effect of microwave time on myosin is not substantially different. The correlation between particle size distribution and protein aggregation was also studied, in terms of time savings, the MWH of short microwave action is preferable since it not only promotes myosin aggregation but also avoids the drawbacks of a rapid warming rate. These discoveries give a theoretical foundation for understanding silver carp myosin under microwave modification, which is critical in the food industry.

## 1. Introduction

China is the leading fish producer, with total fish output expected to reach 27,613,600 tonnes in 2020, with silver carp accounting for 3,812,899 tonnes, second only to grass carp. Furthermore, in 2020, the value of fishery production reached 135,724 million yuan [1]. Surimi products have grown in popularity as the aquatic goods business has grown, owing to its high nutritional content, numerous flavor variants, and portable consumption techniques. The silver carp (*Hypophthalmichthys molitrix*) is a white, high-quality, and reasonably priced fish. Despite its massive output, the earthy flavor and large number of small spines present in the product hinder sales. As a result, turning silver carp into surimi products can improve not only its utilization but also its economic value [2,3]. Improving surimi product productivity and gel quality has been a major focus of research in the manufacture and processing of surimi goods. The proteins in surimi are generally divided into three categories: salt-soluble proteins, water-soluble proteins, and insoluble proteins [4]. The proteins that are soluble in neutral salt solutions and form an elastic gel after heating are mainly salt-soluble proteins, such as myofibrillar proteins, which are composed of myosin, actin, and actinomycin and are the main components of the elastic gel formed by surimi [5]. The gelation properties of surimi directly affect the quality of surimi products, such as color, shape, and taste. Change in protein conformation, as a crucial biomolecule controlling surimi product quality, necessarily impacts the distribution and density of amino acids, as well as the number of hydrophobic cavities on their surfaces, resulting in changes in surface hydrophobicity [6,7]. Surimi’s gel characteristics are regulated not only by endogenous proteases’ degradation and cross-linking by cross-linking enzymes [8], but also by the protein’s conformation and temperature properties. Protein denaturation and hence aggregation can be caused by conformational changes, and correct aggregation is a prerequisite for the creation of gel network architectures [9]. Heat treatment is required for the processing of surimi and, later, for the autoclaving process, and there are several methods for doing so.

Microwave technology has gained popularity in food processing due to its environmental friendliness, safety, and non-toxicity. Rapid microwave heating has been shown to improve the solubility [10,11], gelability [12], emulsification [13], and foaming [11] of food proteins. Rapid microwave heating and high-frequency electromagnetic radiation can cause changes in the structure and aggregation behavior of myosin [14], and microwave heating can cause uneven temperature distribution of surimi products, usually with hot and cool regions [15]. Surimi myofibrillar proteins can only form fully in a low-temperature water bath before being cross-linked in a fast microwave heating process [16]. The rapid warming of MWH and the high frequency effect of electromagnetic waves lead to changes in the structure and aggregation behavior of myosin [17]. However, it can cause uneven temperature distribution of minced fish. Some studies have found that the use of microwave instead of traditional water bath heating in the high temperature section or low temperature section can improve surimi products, especially in terms of improving gel strength and water holding power [18]. The group’s previous research found that that using microwave-assisted water bath heating in the low temperature section and microwave heating in the high temperature section can improve heating efficiency and reduce cooking losses, whereas using water bath heating in the low temperature section and microwave-assisted water bath heating in the high temperature section has relatively good textural characteristics of surimi, retaining more water in the three-dimensional network structure [19]. However, no studies have been done on the impact of the two-stage heating method on the structure and function of surimi proteins.

As a result, different groups of heating conditions for silver carp myosin (SCM) were established in this study. During different heat-induced gel formations, the conformational changes, denaturation, aggregation properties, and differences in gel formation of silver carp myosin were examined. The goal is to reveal the surimi of silver carp gel formation mechanism and to provide theoretical support for the comprehensive elucidation of the functional properties of microwave-modified surimi proteins in the food industry, as well as to provide a certain theoretical research basis for the development of high-quality surimi gel products.

## 2. Materials and Methods

### 2.1. Materials and Reagents

Fresh silver carps (*Hypophthalmichthys molitrix*) were acquired and transported to the lab from a local fish market in Dalian, Liaoning Province, China. After removing the head, scales, and viscera, the fish were rinsed thoroughly in 4 °C deionized water and set away for later use. All chemicals used were of analytical grade except for potassium bromide which is spectrally pure; they were purchased from Sinopharm Chemical Reagent Co., Ltd. (Shanghai, China) or Sigma-Aldrich (St. Louis, MO, USA).

### 2.2. Preparation of Myosin from Silver Carps (SCM)

With minor adjustments, the extraction process of SCM was based on that of Cao et al. [20]. To avoid protein denaturation and hydrolysis, all stages were carried out at a temperature of 0–4 °C. The white meat on the back of silver carp was picked manually, cut into minced meat, and mixed with a 10-fold volume of solution A (0.1 mol/L KCl, 0.02 mol/L Tris, pH 7.0), and homogenized at 3000 rpm then centrifuged at 3000× *g* for 5 min (HR/t20MM vertical high-speed freezing centrifuge, Hunan Hexi Instrument Equipment Co., Ltd., Hunan, China). The precipitate was resuspended in five volumes of solution B (0.45 mol/L NaCl, 0.2 mol/L Mg(CH_3_COO)_2_, 0.001 mol/L EDTA, 0.005 mmol/L-mercaptoethanol, and 0.02 mol/L Tris-HCl solution (pH 6.8)), and the final concentration was adjusted to 0.005 mol/L using ATP. The mixed solution was kept for 2 h at 4 °C. After centrifuging the solution at 8000× *g* for 20 min, the supernatant was diluted three times with solution C (0.5 mol/L KCl, 0.02 mol/L Tris, 0.005 mol/L β-mercaptoethanol, pH 7.5) and stored for 15 min. The mixture was then centrifuged for 15 min at 9000× *g*, and the supernatant was discarded. To the precipitate, a 2.5-fold volume of solution D (0.001 mol/L KHCO_3_) was added, followed by a gradual overnight dilution in a 2.5-fold volume of 0.01 mol/L MgCl_2_. After centrifugation at 10,000× *g* for 15 min, a fold volume of solution E (0.5 mol/L KCl, 0.02 mol/L Tris, pH 7.0) was added to the precipitate, spun at 5000× *g* for 10 min, and the supernatant was gotten and dialysed overnight, it was recognized as myosin. Ultimately, using bovine serum albumin (BSA) as a standard, the protein content was determined using the biuret method [21]. The concentrations of myosin were adjusted to 10.0 mg/mL using 0.5 mol/L KCl (20 mmol/L Tris-HCl, pH 7.5) buffer for later use.

### 2.3. Heating up Methods

In a 50 mL centrifuge tube, 25 mL of 10 mg/mL myosin solution was placed, and two heating stage methods were used, as shown in Table 1. The untreated myosin (control) is a sample that has not been treated in any way. A handheld electronic thermometer was used to measure the temperature in real time (8807, Deli Group Ltd., Ningbo, China). The low temperature section was evenly treated with WH (40 °C, 30 min), whereas the high temperature section was warmed to 90 °C with WH, MH, or MWH, respectively, and the time was recorded. WH: The centrifuge tube was placed in a 90 °C water bath (HH-6, Guohua Electric Co., Ltd., Changzhou, China). MH: The centrifuge tubes were heated at intervals in the center of a microwave oven (G80D23CSL-Q6, Grandis Microwave Oven Electric Co., Ltd., Foshan, China) (10 s heating and 10 s stopping). MWH: Three control experiments using different volumes of deionized water and beakers were set up. MWH1: The centrifuge tube was placed in a 150 mL beaker containing 90 mL of deionized water. MWH2 was placed in a 200 mL beaker containing 100 mL of deionized water. MWH3 was placed in a 250 mL beaker containing 125 mL of deionized water. The microwave was heated for 5 s, followed by a 5 s pause. At 2450 MHz, the microwave output was 800 W.

### 2.4. Determination of Total Sulfhydryl (TS), Active Sulfhydryl (AS), and Disulfide Bond Content

Total sulfhydryl (SH) content was measured according to the method of [22] with a slight modification; 2 mL of 0.2 M phosphate buffer solution (PBS) at pH 8 was added to 0.5 mL of 1 mg/mL SCM solution, followed by 100 μL of 5,5′-Dithiobis-2-nitrobenzoic acid (DTNB) (10 mmol/L DTNB, pH 7.2), mixed well, and then placed in a 40 °C water bath for 25 min. As for active sulfhydryl, 9 mL Tris HCl (0.2 mol/L Tris, pH 6.8, 10 mmol/L EDTA) buffer was added to 1 mL SCM for active sulfhydryl. Four milliliters of the aforesaid mixture were utilized, followed by 0.4 mL of 0.1% DTNB solution and 1 h of reaction time at 4 °C. Meanwhile, a blank was made with PBS, and the absorbance was measured at 412 nm using a SynergyH1/H1M ELISA (Burton Instruments USA, Inc., Gainesville, VA, USA). The disulfide bond content is calculated by subtracting the active sulfhydryl content from the total sulfhydryl content. Both the total sulfhydryl and active sulfhydryl content calculations are based on the following formula:
$$\mathrm{Sulfhydryl}\;\mathrm{content}\;\left(\mathrm{mol}/10^{5}g\right)=10^{5}\times A\times D/13600C$$

Note: *C* is protein concentration, *A* is the SCM solution absorbance value—blank control absorbance value, and *D* is dilution time.

### 2.5. Surface Hydrophobicity

The surface hydrophobicity of SCM was performed as was done previously following the method of Yongsawatdigul and Sinsuwan [23] with modifications. SCM were dissolved in buffer as above to generate a series of concentrations ranging from 0.1 to 0.5 mg/mL, respectively. Two milliliters of protein solution at various concentrations was added to 10 μL of pH 7.5 ANS, mixed thoroughly, and set aside for 10 min away from light. The Multi-Mode Microplate Reader’s parameters were as follows: The excitation wavelength was 303 nm, while the emission wavelength was 485 nm, with a gain of 50. With protein mass concentration as the horizontal coordinate and fluorescence intensity as the vertical coordinate, the slope of the curve was the surface hydrophobicity index (S0) of SCM.

### 2.6. Tryptophan Fluorescence Spectra

The SCM were diluted using PBS to 1 mg/mL. The fluorescence intensity values under this situation were the peaks in the fluorescence profiles, and the measurement was performed using a fluorescence spectrophotometer (F-2700, Hitachi High-tech science, Inc., Tokyo, Japan) with the excitation light commencement at 300 nm, fluorescence emission at 280 nm, and excitation light termination at 450 nm [24]. The purpose of detecting tryptophan fluorescence spectra is to observe the changes in tryptophan fluorescence intensity of myosin after different heat treatments and to clarify the trends in protein conformation by comparison with blank samples.

### 2.7. Ultraviolet (UV) Spectroscopy

UV spectroscopic experiments were performed with modifications to a previously reported approach [25]. With 0.5 M NaCl–20 mM Tris-HCl (pH 7.0), the myosin solution was diluted to 0.5 mg/mL. UV spectra were collected at a scan rate of 10 nm/s in the wavelength range of 190–600 nm. Finally, using Origin 2018, second-derivative spectra (d2A/d2) of the UV spectrum vs. scan wavelength were obtained. UV spectroscopy focused on changes in the molecular conformation of myosin, where the molecular conformation unfolding may reduce the UV absorption and vice versa. The second order derivative and the change in r-value were also used to show the extent to which the different treatments affect the tertiary structure of the protein and thus its gel structure.

### 2.8. Fourier Transform Infrared Spectroscopy (FT-IR)

Slightly modified according to the research method of Liu et al [26]. The SCM powder was completely mixed with KBr at a ratio of 1:50. The scanning wavelength range from 450–4000 cm^−1^. A baseline correction was done at 1600–1700 cm^−1^ using Peakfit Version 4.12 software, and a Gauss peak splitting fit was applied to the second order derivative spectrum to quantify the relative fraction of each secondary structure component based on the integrated area. The α-helix, β-sheet, β-turn, and random coil involved in the gelation process were quantified in this investigation.

### 2.9. Dodecyl Sulfate Polyacrylamide Gel Electrophoresis (SDS–PAGE)

To assess the level and content of protein in supernatant, the SDS-PAGE technique was used with a Bio-Red mini-gel slab electrophoresis equipment (Bio-Rad Laboratories, Inc.). This was based on the previous work done by Liu et al. [27]. Pre-cast TGX gels with a 4–20% gradient were used to examine protein solutions. The samples were diluted to a protein concentration of 5 mg/mL in buffer. Forty microliters of SCM was mixed with 10 μL of buffer (0.25 mol/L Tris-HCl pH 6.8, 10% sodium dodecyl sulfate, 0.5% bromophenol blue, 50% glycerol, 5% polyethanol) and cooked for 5 min in boiling water. After cooling the sample to room temperature, it was centrifuged (2000× *g* for 5 min). The voltage was set to 80 V, the current was set to 30 A, and the timer was set to 300 min. The electrophoresis was started, paused when the strip reached separator gel, set to 120 V, current 30 A, and ended when the strip reached the separator gel’s bottom. After staining with Comas Brilliant Blue R250, the strips were decolored with decolorization solution, and imaged using a Biorad GS900 gel scanner (Bio-Rad, Hercules, CA, USA).

### 2.10. Atomic Force Microscopy (AFM)

The concentration of SCM samples was changed to 1 mg/mL. As a control, unheated SCM samples were used. To generate a myosin adsorption layer, SCM solution was dripped onto the surface of a freshly peeled mica sheet and allowed to dry naturally on a super clean table at room temperature. A TAP 150 probe model (MPP-12100-10) and molecular force operation mode were used for atomic force microscopy [28]. T = 1.5–2.5 m, L = 115–135 m, and W = 25–35 m were the micro-cantilever thicknesses; the resonance frequency (f_0_) = 150–200 KHz; the elastic coefficient (k) = 5 N/m; and the resolution = 256 × 256. The method allowed visualization of the surface morphology and aggregation of myosin, and used the average roughness to reflect the degree of aggregation of the protein.

### 2.11. Scanning Electron Microscopy (SEM)

The Hitachi SU1510 (Hitachi Ltd., Tokyo, Japan) was used for SEM characterization with a 15 kV accelerating voltage. Powders (freeze-dried SCM) were conductively connected to an electron microscope sample stage using conducting carbon tape and sputter-coated with platinum/gold before observation. It is possible to observe the regularity of the sample structure, the roughness, and the size of the pores. These indicators have a direct influence on the gelation behavior of surimi.

### 2.12. Particle Size Distribution

The particle size distribution of SCM was measured by dynamic light scattering (DLS) using ZS 90 equipped with 4 MW He Ne ion laser (L = 633 nm). Place 1 mg/mL myosin solution in a 1 cm diameter quartz tube, and the detection angle is 90° at 25 ± 0.1 °C.

### 2.13. Statistical Analysis

Origin 2019 software is utilized to calculate the second derivative of UV scanning spectral data, and Microsoft Excel 2010 is used for data processing and drawing. Duncan used IBM SPSS statistics 21.0 to conduct significance analysis and correlation analysis. *p* < 0.05 revealed significant differences between lowercase letters.

## 3. Results and Discussion

### 3.1. Sulfhydryl and Disulfide Bonds

The interaction of polar groups between proteins in the presence of heat treatment or electromagnetic waves leads to the exposure of sulfhydryl groups in protein molecules, some of which recombine to form disulfide bonds to maintain the three-dimensional structure of surimi gels. Therefore, characterization of TS and AS is necessary to understand the elasticity of surimi gels. Figure 1 shows that the TS content after heating is higher than the blank, although it is not significantly different from the blank (*p* > 0.05). MH has a lower TS and a higher AS than WH, as well as a higher disulfide bond content. With the extension of microwave action time, the AS of MWH increased and the content of disulfide bond decreased. The force of polar groups between proteins exposed by an electromagnetic field exposes sulfhydryl groups concealed in protein molecules [29,30], some of which can rebuild disulfide bonds through an oxidation reaction [31]. Surimi gel’s three-dimensional conformational stability is maintained by a covalent link of disulfide bonds. The covalent crosslinking increases as the quantity of disulfide bonds increases, improving the elasticity of surimi gel [32]. Electromagnetic fields expose the sulfhydryl groups hidden in the protein molecules, and some of these sulfhydryl groups reform disulfide bonds. From Figure 1 we can observe that the difference in heat treatment did not have a significant effect on the TS and AS, probably due to the large experimental error and probably because this indicator does not give a clear picture of the effect of the treatment on the protein molecules of the surimi.

### 3.2. Surface Hydrophobicity (S_0_-ANS)

Myosin molecules unfold when heated, exposing the hydrophobic amino acid residues within the protein, and leading to an increase in surface hydrophobicity. Hydrophobic interactions are a non-covalent force that maintain the gel structure and the characteristics of amino acid residues in myogenic fibrils directly influence the properties of the product. The surface hydrophobicity (S_0_-ANS) of SCM revealed diverse outcomes under different heat treatments, as illustrated in Figure 2. The S_0_-ANS of SCM is low after MH treatment, indicating that the degree of protein denaturation is low and the degree of protein structural expansion is poor [33,34]. It is possible that the microwave heating rate is too fast, achieving the same end temperature, and the overall heat received by myosin is insufficient (the integral of temperature and time), leaving insufficient time for myosin hydrophobic residues to expand. However, because WH is greater than MH, the degree of conformational change of MH myosin is less than that of WH, thus MH may block protein conformational change [15]. Most notably, MWH altered myosin’s conformation, extended the helical structure, exposed more hydrophobic amino acids, and when paired with a fluorescent probe, increased the hydrophobicity index [35,36]. The S_0_-ANS reduced and then increased as the microwave action time in MWH was increased, indicating that myosin begins to unfold, revealing hidden nonpolar amino acids and facilitating protein denaturation. Because of the increased hydrophobic interaction, myosin begins to aggregate and form polymers when the concealed hydrophobic amino acids are exposed to their maximum level, and the hydrophobic interaction between molecules may be the driving factor behind surimi gel production [15,33]. Although there is no significant difference between WH and MWH in terms of surface hydrophobicity, this does not indicate that MWH does not have an advantage. In terms of time, MWH takes less time than WH, and it is not always comprehensive to judge from just one characterization.

### 3.3. Tryptophan Fluorescence Spectra

In comparison to the control, the tryptophan fluorescence intensity of SCM rose and was red-shifted after heating, as seen in Figure 3. These results suggest that the fluorescent amino acids were exposed to a hydrophilic environment, causing a change in protein structure. The intensity of SCM tryptophan fluorescence after MH treatment was higher than WH, while MWH was higher than MH and WH. The intensity of tryptophan fluorescence increased with increasing microwave action time in the case of MWH, possibly because the microwave-treated myosin unfolded to a lesser extent than WH, or because myosin aggregation under rapid microwave heating may result in some tryptophan residues being encapsulated inside the myosin molecule [37,38]. More tryptophan residues and hydrophobic groups would be exposed outside the myosin molecule, resulting in increased protein fluorescence intensity. Although tryptophan is a hydrophobic amino acid, the intensity of tryptophan does not necessarily follow the same trend as surface hydrophobicity. A comparison of Figure 2 and Figure 3 shows that the microwave water bath treated surimi showed consistent trends in these two characterization indicators but the blank control, WH and MH, showed opposite trends.

### 3.4. UV Scanning Spectroscopy

The UV absorption peak of SCM is around 265 nm, indicating the presence of aromatic amino acids, as seen in Figure 4A. The aromatic chromophores (Phe, Tyr, and Trp residues) of proteins can be observed in the near UV region at 250–320 nm and this has been widely used to assess changes in the tertiary structure of proteins during processing [39]. The control group’s absorbance was substantially higher than the heated groups. The higher absorbance of MWH compared to MH could be due to the unfolding of the molecular conformation of SCM caused by microwaves, or it could be due to the encapsulation of exposed tryptophan residues in the SCM aggregates of MH, resulting in lower UV absorption, whereas MWH could be able to slow down the unfolding of the molecular conformation of myosin [13,35]. Heating causes a slight blue shift in the UV absorption peak (Figure 4B), which is caused by a conformational change that exposes the tyrosine and tryptophan to the protein surface, creating a polarity shift in the microenvironment where they are located [40]. The second order derivative spectrum for this sample of silver carp has two positive peaks and two negative peaks between 280 and 300 nm, as evidenced by the relevant findings of our research group [41]. The absorption spectra of the second order derivatives show two maxima and two minima at 288 nm and 295 nm, and 288 nm and 291 nm, respectively, where the peak at 288 nm is caused by the combined action of tryptophan and tyrosine, and the peak at 295 nm is caused by tryptophan alone [42]. It can be seen from Figure 4C that the r values of SCM all decreased after heating. Except for WH and MWH2, the r values of the other modalities were significantly lower (*p* < 0.05), indicating that MH, MWH1 and MWH3 have a greater effect on the tertiary structure of myosin, promoting SCM aggregation and making the formation of gel structure more favorable, and this result is consistent with the tryptophan fluorescence spectral results (Figure 3).

### 3.5. FT-IR Spectra

FT-IR spectroscopy is useful for studying the conformational changes of myosin in various environments. The amide I band (1700–600 cm^−1^) is useful for gaining insight into the secondary structure of proteins. In this study, only the α-helical and β-sheet secondary structure components, which are the main structures involved in the gelation process, were quantified, and the subpeaks corresponded to the secondary structures as follows: 1610–1639 cm^−1^ was considered to be the β-sheet structure and 1661–1680 cm^−1^ was considered to be the α-helical structure [7]. Figure 5A shows the exact percentage of each secondary structure in myosin for the different treatments and Figure 5B shows the FT-IR data for myosin samples that have undergone different heat treatments. The α-helix content of heated SCM decreased compared to the control (*p* > 0.05), with a significant decrease in MWH2 (*p* < 0.05). The α-helix content of a protein plays a crucial impact in its gelation, with a lower α-helix content favoring myosin aggregation to form a gel with a denser reticular structure [43]. Thermal gelation resulted in a shift from α-helix to β-sheet, possibly due to denaturing aggregation of the protein structure at high temperatures (90 °C) [44]. The β-sheet content of heated myosin increased compared to the control, with MH, MWH1, and MWH3 having significantly higher β-fold content than the control (*p <* 0.05). The increase in β-sheet indicates that SCM has a more stable structure compared to the control. According to Lee et al. [45], hydrogen bonding makes it easier to create more β-sheet structures within protein aggregates. The β-sheet had better stability and mechanical properties than gels with a high α-helix concentration.

### 3.6. SDS-PAGE

Myosin heavy chains (MHC 200 kDa), helical tail of myosin (LMM 70 kDa), actin (43 kDa), tropomyosin (TM, 38 kDa), and myosin light chains (MLC, 20 kDa) are all well-known components of myosin [46,47]. The results (Figure 6) showed that no new bands developed when compared to unheated SCM, although protein content changed, myosin band brightness and width decreased, notably MHC (200 kDa). The MHC band strength of myosin decreased after heating, which was a critical protein in the creation of cross-linked network structure. The decrease in MHC band in MWH with increasing microwave action time is not clearly seen in Figure 6. However, the decrease in MHC intensity in myosin due to heat treatment is clearly observable. MHC was easily cross-linked by chemical bonds following heat treatment due to the expansion and crosslinking of the protein during heating, resulting in a lower stripe strength on the SDS-PAGE gel, where the WH band strength fell and MH decreased most noticeably. The MHC band strength in MWH declines as the microwave action time increases, although MWH slows down the reduction of MHC band strength compared to MH. Microwave, according to Liang et al. [48], can significantly decrease hydrolysis and speed protease inactivation due to its rapid heating rate. Although LMM did not totally vanish after heating, the brightness of the myosin band was greatly reduced, indicating that microwave treatment encouraged myosin expansion. Microwave-induced covalent bonds contribute to protein structural stability, hence increasing gel strength [15]. In short, the largest alterations in the electrophoretic map occur mostly in MHC, whereas the actin bands show essentially no change. The electrophoretic results of Tornberg et al. [49] revealed that the actin bands fade later than the myosin bands after heating. Actin is resistant to protein hydrolysis, according to Rawdkuen et al. [50], hence the actin bands are unaffected.

### 3.7. Effect of Heat Treatment on the Surface Morphology of SCM

Figure 7 shows the surface morphology and aggregation phenomena of unheated and differently heated SCM in 2D and 3D pictures and cross-sections, where the average roughness reflects the degree of protein aggregation [51]. Untreated SCM is uniformly adsorbed on the mica surface, primarily in the form of monomers, fibrillated soluble oligomers, and protofibrils. Thermally induced aggregation can form soluble and insoluble particles [52]. Myosin aggregates of WH and MH produced small clustered aggregates uniformly scattered on the mica surface, but myosin aggregates of MWH were more cross-linked with each other and formed bigger clustered aggregates unevenly distributed on the mica surface. As observed in the cross-sectional picture, heating disturbed the structure of myosin and lowered its cross-sectional height. The varied heating techniques reduced the roughness when compared to the control (4.91 ± 0.77 nm), showing that myosin aggregation was encouraged to some extent. The MWH promoted myosin aggregation more effectively, although mean roughness of myosin did not differ substantially among the five heating techniques (*p* > 0.05).

### 3.8. SEM

The microstructure of the gel has a significant impact on the physical properties of gel textures. Microstructural analysis should provide useful information about the structural evolution of gel formation. The surface of the unheated SCM is reasonably smooth, as seen in Figure 8. Because of the random aggregation of myosin unfolding, the heated SCM has an uneven, rough, and large pore size structure, generating a disordered gel network [53]. The pore size of WH’s myosin network structure is big, and this irregular gel structure with large pores is linked to surimi’s poor gel performance [54]. Surimi cooked in a traditional water bath had a coarser microstructure, resulting in a worse texture and weaker gel strength. Rapid microwave heating has a thermal effect that expands the protein and improves the microstructure of the gel, resulting in a dense mesh structure [16]. The number of dense aggregates increased significantly as the microwave heating time was increased, and the number of small pores increased with uniform distribution, suggesting that the gel structure formation is easily affected by the rate of protein unfolding or cross-linking, and that when the unfolding rate is faster than the cross-linking rate, a uniform and dense gel microstructure is formed.

### 3.9. Particle Size Distribution

DLS is a quantitative, sensitive, and effective approach for tracking protein aggregation development [55,56]. The particle size distribution of SCM treated with control and other heating methods has a single narrow peak, whereas the size distribution of untreated SCM is greater, indicating that the original myosin maintains a highly aggregated structure, as seen in Figure 9A. The particle size of SCM reduces and the curve peak changes to the left after treatment with various heating methods, which is due to denaturation of protein molecules and non-covalent link breakage [57]. The particle size of WH-formed myosin clusters is considerably larger than that of MH and MWH when heated to the same temperature of 90 °C. It is possible that myosin has more time to grow and aggregate throughout the heating process because WH heats up slowly. Particle size peak can be used to assess the particle size intensity distribution. The narrower the particle size distribution, the more uniform the particle size [58], and the larger the particle size peak, the smaller the particle size distribution. WH has a greater particle size peak than MH and MWH, indicating that most of the myosin particle sizes in MH and MWH are on a tiny scale. Figure 9B shows that WH has a substantially larger average particle size than MH and MWH (*p* < 0.05). The increase in average particle size could be due to an increase in intermolecular covalent bonds and hydrophobic interactions, which is consistent with the finding that myosin heated in the microwave has a smaller average size than myosin heated in the water bath [38,59]. MWH has a higher mean particle size than MH, and the increase in particle size is attributable to the development of aggregates induced by disulfide bonds and hydrophobic contacts between myosin molecules, similar with the results of sulfhydryl and disulfide bonds (Figure 1). The results of AFM showed that MH and MWH could also accelerate myosin aggregation and facilitate the creation of gel formations (Figure 7).

### 3.10. Correlation Analysis of Protein Aggregation Behavior and Particle Size Distribution

Table 2 shows the relationship between SCM aggregation behavior (surface hydrophobicity, total sulfhydryl content, reactive sulfhydryl content, and disulfide bond content) and average particle size. With correlation values of 0.247 and 0.862, the average particle size of SCM was favorably linked with surface hydrophobicity and reactive sulfhydryl concentration. SCM occurs as a monomer when not heated. When denatured and aggregated, the particle size increases, and the free myosin molecule’s head will form a disulfide connection with the tail like a bridge [15]. Hydrogen bonds, disulfide bonds, electrostatic contacts, and hydrophobic interactions combine unfolded protein structural domains to form a three-dimensional network [60]. Heat causes myosin to unfold, exposing hydrophobic residues and causing myosin head-heads to cross-link and aggregate. Myosin tails denature at the same time to produce more molecular cross-links, allowing gel networks, clusters, and protein aggregates to form through hydrophobic contacts, etc., resulting in an increase in myosin particle size [61]. The total sulfhydryl concentration and disulfide bond content of myosin were negatively associated, with correlation coefficients of −0.06 and −0.781, respectively. Cross-linking of disulfide bonds in the head region of myosin caused myosin to aggregate, increasing the average particle size of myosin and forming larger protein aggregates. The decrease in total sulfhydryl content and cross-linking between disulfide bonds caused myosin to aggregate, increasing the average particle size of myosin and forming larger protein aggregates [15,62].

## 4. Conclusions

On the structure and function of SCM, the effects of normal water bath heating, microwave heating, and microwave-assisted water bath heating were explored. After MWH treatment, SCM displayed increased sulfhydryl groups as well as surface hydrophobicity. Under fast MWH, SCM produced a dense reticular structure, as seen by SEM. As the microwave time was increased, the distribution of tiny pores became more uniform. Finally, MH and MWH can aid in the creation of gel structure by promoting myosin aggregation. However, the effect of MWH time on myosin was not significantly different, and when considering the time cost, a short time MWH is preferable, not only to promote myosin aggregation, but also to prevent the drawbacks of a too-fast heating rate. These findings may aid in a better understanding of SCM gel formation in a microwave field, as well as improving the quality of myosin-based foods through microwave treatment. To give further data support for future large-scale production, more studies involving computer simulation and other precise temperature control technologies are needed.

## Figures and Tables

**Figure 1 foods-11-01071-f001:**
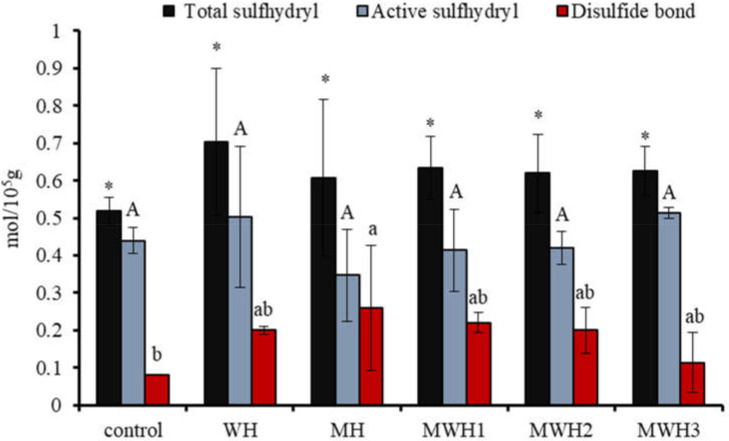
Changes in the content of myosin sulfhydryl groups and disulfide bonds under different heating conditions. Values with different upper-case letters, lower-case letters, and * are significantly different (*p <* 0.05).

**Figure 2 foods-11-01071-f002:**
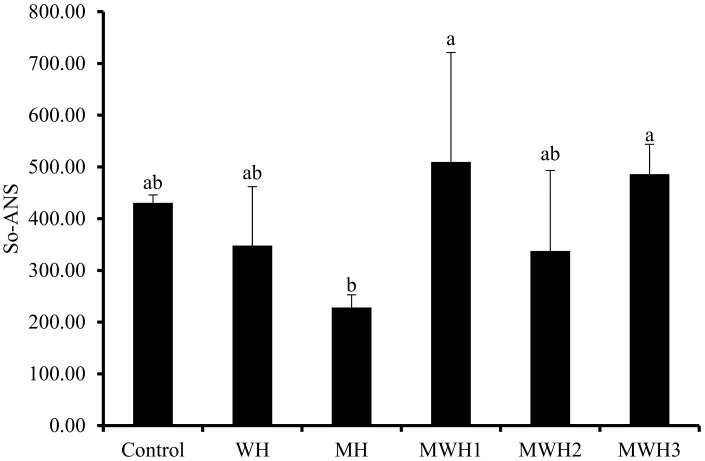
Changes in surface hydrophobicity under different heating conditions. Values with different lower-case letters are significantly different (*p <* 0.05).

**Figure 3 foods-11-01071-f003:**
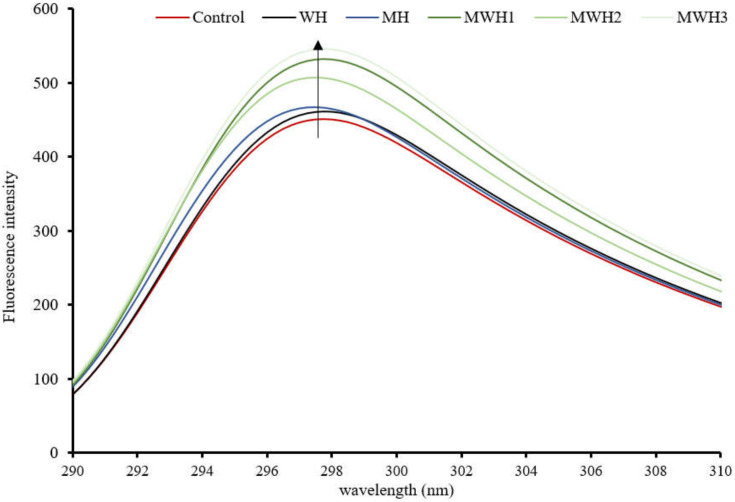
Tryptophan fluorescence spectra of SCM under different heating methods.

**Figure 4 foods-11-01071-f004:**
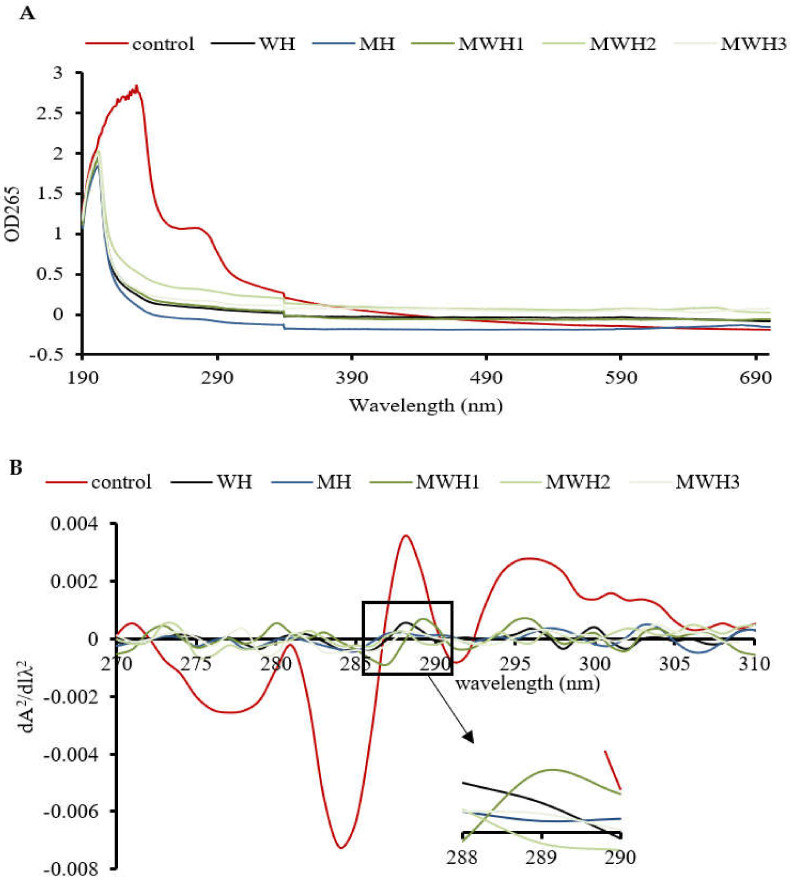
(**A**) The UV absorption spectra of SCM at different heating methods; (**B**) the second derivative spectra of myosin on UV absorption spectra at different heating; (**C**) the ratio of the peak-to-trough values for the two main peaks. Values with different lower-case letters are significantly different (*p <* 0.05).

**Figure 5 foods-11-01071-f005:**
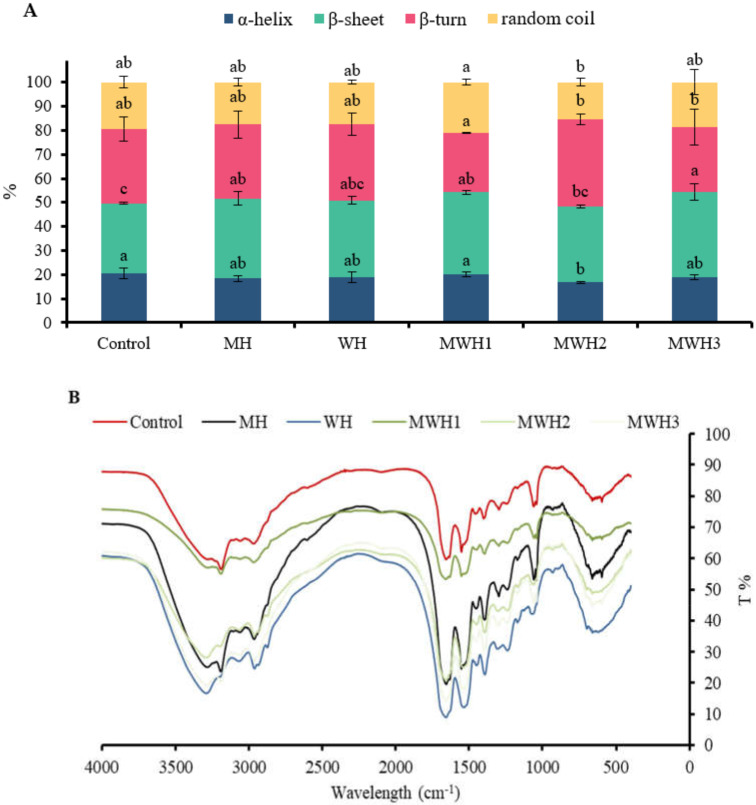
(**A**) Changes in the secondary structure of myosin under different heating conditions. Values with different lower-case letters (a–c) are significantly different (*p <* 0.05). (**B**) Fourier transform infrared spectroscopy of SCM with different heating methods.

**Figure 6 foods-11-01071-f006:**
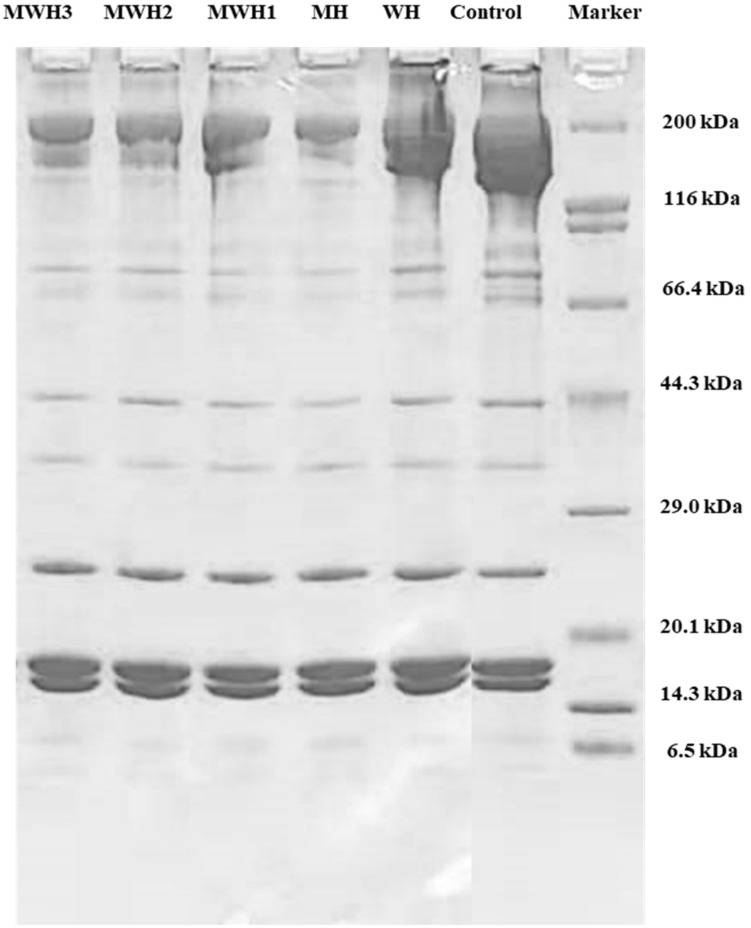
SDS-PAGE patterns of SCM heated by different methods.

**Figure 7 foods-11-01071-f007:**
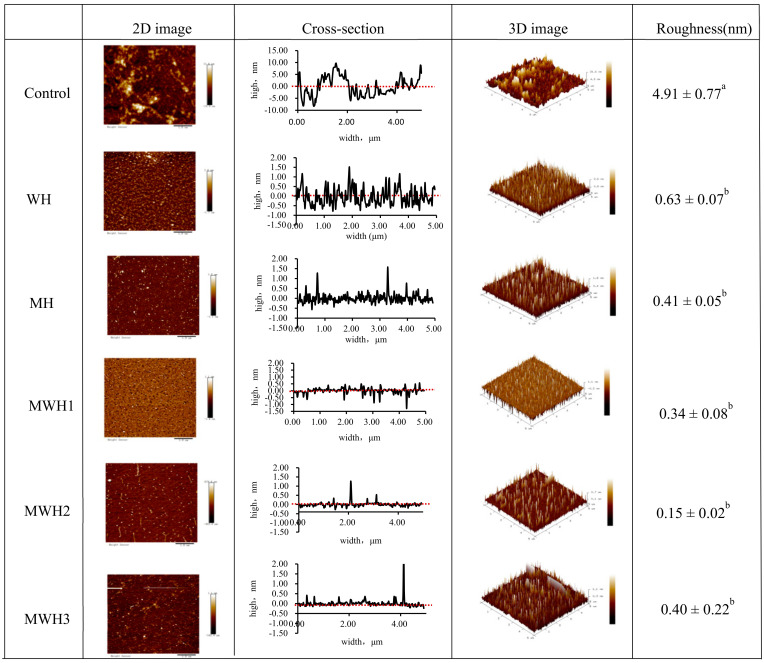
2D and 3D images of AFM with different heating methods of SCM and the cross-section reflected the largest roughness on the 2D images. Different lowercase letters indicate significant differences in mean roughness values between SCM in different heating modes (*p <* 0.05).

**Figure 8 foods-11-01071-f008:**
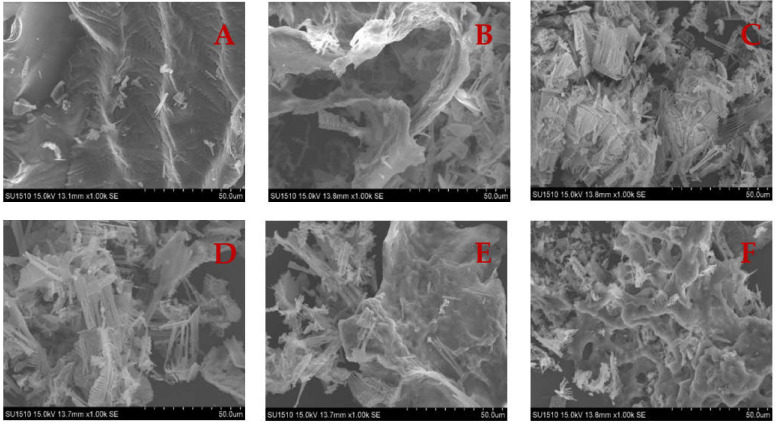
Microstructure of SCM under different heating conditions. (**A**) Control; (**B**) WH; (**C**) MH; (**D**) MWH1; (**E**) MWH2; (**F**) MWH3.

**Figure 9 foods-11-01071-f009:**
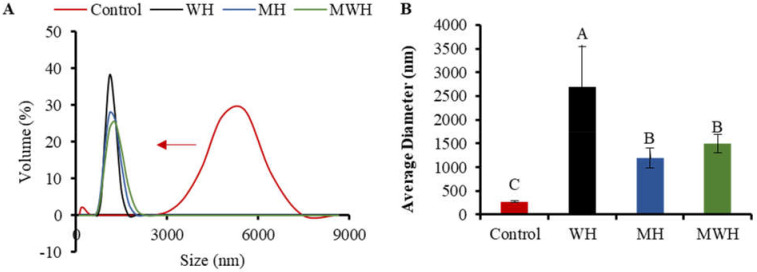
(**A**) The results about particle size distribution of SCM under different heat treatment. (**B**) Average particle size distribution of SCM in different heating modes. Values with different upper-case letters are significantly different (*p <* 0.05).

**Table 1 foods-11-01071-t001:** Different heating methods of myosin.

Group	Heating Methods
Control	--
WH	40 °C 30 min + WH 15 min 11 s
MH	40 °C 30 min + MH 1 min 06 s
MWH1	40 °C 30 min + MWH 4 min 45 s
MWH2	40 °C 30 min+ MWH 5 min 10 s
MWH3	40 °C 30 min+ MWH 9 min 10 s

Note: min means minutes, s means seconds.

**Table 2 foods-11-01071-t002:** Correlation analysis between protein aggregation behavior and particle size.

Indicators	Surface Hydrophobicity (S_0_)	Total Sulfhydryl	Active Sulfhydryl	Disulfide Bonds
Average particle size	0.247	−0.06	0.862	−0.781

## Data Availability

Data is contained within the article. More detailed data generated by this study is available on request from the corresponding author.

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
