# Peer review of "Changes in Myosin from Silver Carp (Hypophthalmichthys molitrix) under Microwave-Assisted Water Bath Heating on a Multiscale"

_foods, 2022, doi:10.3390/foods11081071_

Round 1

Reviewer 1 Report

The present manuscript deals with the changes in myosin from silver carp under microwave-assisted water bath heating on a multiscale. The investigation is interesting ad has scientific merit.

My suggestions to the authors that they may confirm their findings with some different methods, like Differential  Scanning Calorimetry. Which is useful method useful technique for studying changes in myofibrillar protein.

Author Response

Response: We are grateful for your encouraging and pertinent comments on our paper. The method like Differential Scanning Calorimetry you mention is nice, but the characterization presented in the article already allows changes in myofibrillar proteins to be observed. We tried DSC in previous studies Effects of Heating Methods on the Structure and Physicochemical Properties of Silver Carp Myosin. Different warming methods did not entirely remove the absorption peaks of myosin, showing that the tertiary structure of myosin was not totally denatured and that MWH was able to affect the thermal stability of myosin, with the spatial structure being the most stable (Liang et al., 2021).

Reference:

LIANG Wenwen, YANG Tian, GUO Jian, WANG Qiukan, CONG Haihua, & CHEN Shenjun. Effects of Heating Methods on the Structure and Physicochemical Properties of Silver Carp Myosin. Food Science, 2021, 42(21), 24-31 (in Chinese).

Reviewer 2 Report

In the present research, the content of active and bound sulfhydryl groups after thermal treatments was determined. Surface hydrophobicity through protein denaturation, surface morphology after heat treatment, confirming gel microstructure, and particle size distribution were evaluated to determine structural changes in myosin protein composition. silver carp under different treatment times by application of microwave thermal energy. Through these tests carried out under different times of thermal treatment to that with microwaves, the authors found that the treatment of application of thermal energy by means of microwaves is better in short terms.

I consider that the topic of the paper is relevant and interesting. The manuscript is well written and correctly justified and substantiated, so I have no comments or observations that could contribute to improve the work.

Author Response

We highly appreciate the encouraging comments on our manuscript. We further revised the logic of the article as well as the language.

Reviewer 3 Report

The study looked at the effect of microwave-assisted water bath heating (MWH) on silver carp myosin (SCM) structures. The aim is to understand SCM gel formation in a microwave field so that quality of future myosin-based foods can be controlled. The manuscript requires major attention because it lacks novelty and the quality of the presentation is low. This made the study not appealing and hard to understand.

Major concerns:
Similar work have been published my Li, et al. (2020) https://doi.org/10.1016/j.foodchem.2020.127104 who looked at using two-step microwave heating on the gelation properties of golden threadfin bream myosin and Wang, et al. (2019) https://doi.org/10.1016/j.jfoodeng.2019.04.001 who looked at thermal gelation of Pacific whiting surimi during microwave assisted pasteurisation. The effect of microwave heating on myosin structure has been reported therefore the current study presented in this manuscript did not contribute new knowledge.

The authors should justify the importance of studying myosin in the introduction. Why is gelation important for surimi? What other factors affect the gelling properties in surimi?

In the introduction, where's the evidence to proof that MWH is better than normal MW heating and water bath heating? You only stated the MWH can improve heating efficiency, reduce cooking losses improve surimi texture, etc, but how?

Other comments:

Line numbers are missing from the draft manuscript making it hard to refer to areas that need addressing.

Better wording in the abstract, especially the description of the results. The use of increase and decrease is too excessive. Makes the abstract hard to read.

Introduction, first sentence, the tonnage values are wrong. How can silver carp tonnage be more than the total fish output value in China?

Material and Methods, Preparation of myosin from SCM subheading needs numbering.

"Minced fish muscle hand-picked white meat..." doesn't make sense. Please re-write this sentence.

Table 1: What does those numbers mean? 15'11" for example.

Equation 1 was mentioned twice. Please delete one of them.

What is PBS?

How to calculate the active sulfhydryl? Where's the equation?

Why do you need to measure tryptophan with fluorescence spectra?

What is UV spectroscopy used for?

WHere's the justification for using AFM and SEM?

You need to define the abbreviation TS when it is first used.

What is the significance of finding out the TS and AS content in myosin? 

You mentioned that electromagnetic field exposes sulfhydryl groups concealed in protein molecules. If that is the case, TS and AS for MH should be higher than WH, but your results doesn't show this. Please explain.

How does surface hydrophobicity related to surimi gel production? Is it more desirable for the surimi protein to be more hydrophobic or hydrophilic?

The surface hydrophobicity of WH is not significantly different from MWH, so what is the benefit of MWH?

Under methodology and methods, how many times did you replicated your experiments?

The tryptophan intensity results should mirror the results from surface hydrophobicity because more tryptophan residues and hydrophobic groups are exposed during MWH treatment. But your results showed that this is not true, please explain.

What are the significance of aromatic amino acids? You need reference to confirm whether the SCM absorption peak at around 265 nm us the presence of aromatic amino acids.

In Figure 4, the second derivative spectra of myosin does not look like a true spectra and it is not convincing. It looks more like noise. How can you be so certain it is not noise?

Figure 5B needs more discussion. Which peaks correspond to alpha-helix, beta-sheet, etc?

The MHC band strength in MWH declines as the microwave action time increase... why?

Liang et al. (2020) showed microwave heating can significantly decrease hydrolysis and speed protease inactivation due to its rapid heating rate. How and why does it happen?

MWH promoted myosin aggregation more effectively. Please explain why?

Table 2 is missing units for average particle size.

So is particle larger particle agglomerate more desirable? Figure 9B shows WH particle size are larger than MH and MWH, is this a good thing?

Please proof read your manuscript, there are a lot of errors, too abundant for me to include in this review.

Round 2

Reviewer 3 Report

The authors have satisfactory addressed my comments but they should be included in the revised manuscript. Otherwise, the manuscript is not clear with explanation and lacks in-depth discussion.

I highly recommend that the authors include explanations that were used to addressed my comments  into the manuscript otherwise the manuscript is of low scientific impact.

Author Response

Thank you very much for your comments and I have reworked our draft based on your suggestions. Please see the attachment.
